# Developing Green–Building Design Strategies in the Yangtze River Delta, China through a Coupling Relationship between Geomorphology and Climate

Yuan Zheng [1], Yuan Sun [2,3,]*, Zhu Wang [2] and Feng Liang [1]

1 School of Architecture and Urban-Rural Planning, Fuzhou University, Fuzhou 350108, China
2 College of Civil Engineering and Architecture, Zhejiang University, Hangzhou 310058, China
3 China Institute of Urbanization, Zhejiang University, Hangzhou 310058, China
* Correspondence: 22012137@zju.edu.cn

**Abstract:** Many studies have developed green strategies and technologies for urban construction, but they sometimes ignored the intensive and dynamic relationships between people, buildings and the natural environment. This study focused on how to generate green building design strategies dealing with a coupling relationship between geomorphology and climate, which took an insight into the built environment in a particular locality of the Yangtze River Delta region in China. First, we imported climatic data from six cities into a bioclimatic evaluation tool, named 'Weather Tool', to assess the effectiveness of the existing passive design strategies (passive solar heating, thermal mass effects, exposed mass and night purge ventilation, natural ventilation, direct evaporative cooling and indirect evaporative cooling). Second, we employed the topological method to identify the characters of the vernacular dwellings by interpreting their adaptations to the local topographical and climatic conditions. Consequently, the green building design strategies in the Yangtze River Delta region were developed through the macro, middle and micro levels to examine group patterns, single-building forms and building components in a particular locality. The main findings were shown as follows: (1) the common passive strategies played a role with different effects in the Yangtze River Delta region, which acted as a basis for choosing the most effective strategies; (2) the local dwellings presented a comprehensively sustainable paradigm with architectural prototypes that could be selectively inherited and applied in contemporary design. (3) Those particular strategies, which were evaluated through bioclimatic tools and developed from the vernacular dwellings, gave specific suggestions on green building design in the Yangtze River Delta region, providing approaches for architects and developers to promote more environmentally responsive sustainable development.

**Keywords:** green building design strategy; Yangtze River Delta region; bioclimatic evaluation; architectural prototype; geomorphological adaptation; climate-responsive





## 1. Introduction

### 1.1. Research Background

Urban construction is facing rising threats due to climate change and regional characteristics are being lost through modern urbanization [1]. Developing green buildings is essential for maintaining society in a dynamically changing context of climate and landscape and for achieving a diverse local identity and culture. Accordingly, current studies have sought sustainable and nature-based solutions worldwide from vernacular architecture [2], heritage sites [3], historical landscapes [4], etc. Due to the diverse localities with varying natural factors, researchers have usually focused on particular regions to discuss the provincial sustainability of human settlements [5]. For example, Yahya et al. [6] traced the sustainable architectural practices in the Middle East to create design

approach models for architects and scholars. In addition, Lin et al. [7] structured a research workflow in urban and building aspects to find supportive evidence for vernacular architecture renovation.

Transitioning from conventional to sustainable requires a combination of historical and modern contexts [8]. However, Chen [9] found that due to rapid but low-quality urbanization over the last two decades, some Chinese cities were experiencing a confusing crisis of local identity, where typo-morphological research on urban texture and architecture was relatively rare and superficial. Meanwhile, the rising dependence on high technology and building intelligence gradually has shown a higher possibility of weakening the modern building capacity due to passive self-adaptation [10] and environmental factors, expressing few humanistic and cultural characteristics from architectural prototypes [11].

### 1.2. Human–Land Relationship in Urban Construction

As a primary use of land use, building construction largely reflects the human–land system through complex interactions with diverse environmental elements [12]. The construction system in a particular locality couples with the external environment and human society via its economy, socio-culture, technology, lifestyle and values. Specifically, the environment provides a necessary resource supply and support for human activities but limits and affects the regional architectural construction properties (i.e., settlement organization, urban planning, spatial layout, building envelopes, materials and techniques). Within a certain geographical scope, the self-regulation of the natural ecosystem is confined. If human activities exceed the compensation, self-purification and regulation capacity of the environment, the result, sometimes destructive, will inhibit the construction of regional buildings. Therefore, construction activities should promote the harmony, integrity and sustainability of the human–land relationship (Figure 1).

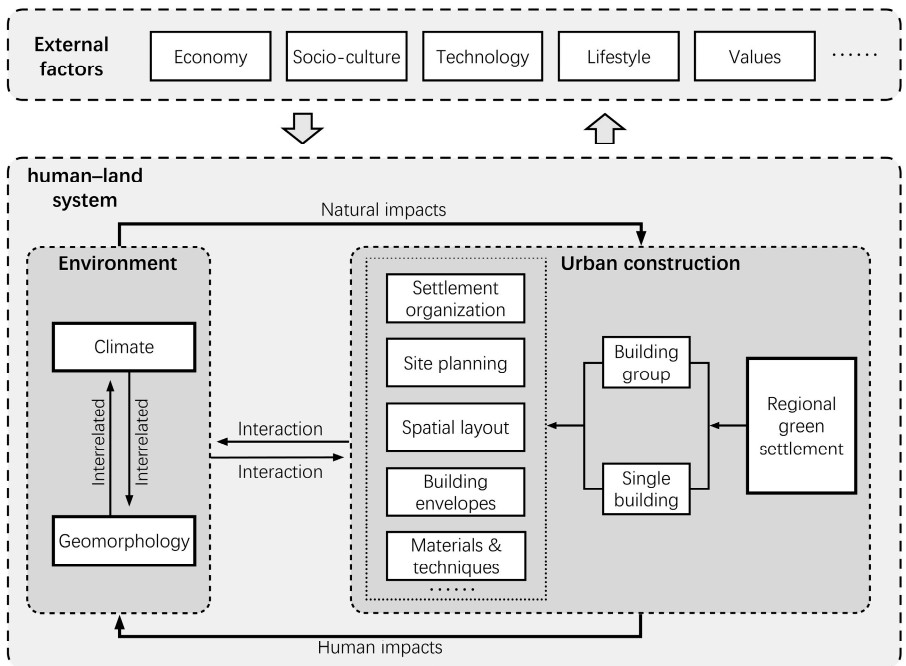

**Figure 1.** Illustration of the human–land relationship in urban construction.

Climate and geomorphology are two interrelated and interacting environmental elements in the natural system. Acting as external triggers, both influence local construction activities and attitudes [13]. Although residents are unable to change the macro-level environments, they can select middle-level surroundings and create a suitable micro-level environment for living [14]. In this way, the deviation between the environmental conditions and human settlement demand could be modified step by step.

*1.3. Generating Green–Building Design Strategy*

Green building refers to healthy facilities that are designed and built in a resource-efficient manner using ecologically-based principles [15]. It involves four aspects: environmental sustainability, life-cycle perspective, the health issues of occupants and the impacts on the community [16]. Green building design, which is the beginning session, deals with the approaches for sustainable construction, indoor thermal comfort, energy efficiency, etc. [17] at the early stage. For example, Makram et al. [18] proposed nature-based design strategies and methods for the development of a comprehensive framework for sustainable architectural design. Wang et al. [19] raised a multi-objective optimization model at the conceptual design stage to assist designers with green building design. Ahmad et al. [20] incorporated various green building strategies and techniques into the design process to achieve thermal comfort using different methodologies. Current studies have raised diverse strategies, including site planning, building form, envelopes, sunlight shading, natural ventilation and thermal mass materials (Figure 2). They have opened a perspective for achieving local adaptation in response to the environmental factors, especially for urban geomorphology and climate.

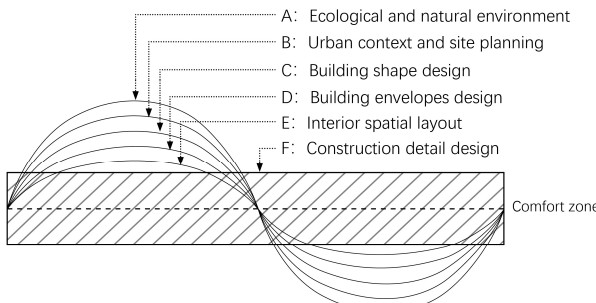

**Figure 2.** Hierarchy of the green building design at different scales.

(1)    Bioclimatic Evaluation

Bioclimatic evaluation is defined as one of the pre-evaluation methods considering the local climate conditions, which allows for a comparative analysis of the outdoor climate conditions and indoor environment performance to assess the effectiveness of the common passive design methods. It helps architects choose effective approaches for enhancing building performance. Olgyay initially raised a systematic theory on bioclimatic design approaches [21]. The relevant analysis approaches include the Givoni bioclimatic charts [22], the Watson design principles [23], the Mahoney bioclimate tables [24] and the Evans comfort triangles [25]. They were widely applied over recent years [26,27]. For example, Dnyandip et al. [28] developed a bioclimatic analysis tool for the evaluation of the cooling potential of passive strategies in different Indian climatic zones. Pajek et al. [29] developed a bioclimatic potential prognosis for 21 characteristic locations in the Alpine-Adriatic region to predict their building energy performance.

(2)    Typology

Local dwellings demonstrate a comprehensively sustainable paradigm using passive design principles that provide comfort, from which many lessons can be learned and applied in contemporary construction practice. These tacit principles could be interpreted as multi-scale passive bioclimatic wisdom, which can also be inherited and applied in contemporary construction practice. The conventional wisdom of construction in a particular locality comes from the people's cognition and response to their living environment, providing reference and experience for contemporary construction strategies. Chandel et al. [30] identified some of the vernacular architectural features that were energy-efficient and raised the opinion of utilizing conventional materials to improve thermal comfort in modern buildings worldwide. Meir et al. [31] gave an opinion on a design approach,

learning from vernacular architecture to optimize contemporary building performance. Manzano-Agugliaro et al. [32] highlighted the use of vernacular typology to promote current bioclimatic architecture as one of the principal scientific research trends.

In this study, we aimed to develop effective green building design strategies that were adaptive to the local geomorphology and climate. We developed a research framework that combined the bioclimatic pre-evaluation and architectural typology to assist in selecting the appropriate strategies. Section 2 details the research methods in two steps. First, evaluating the existing mainstream passive design strategies based on the local climatic data, and second, learning from the past to generate detailed design points by recognizing the architectural prototypes of the vernacular dwellings. Section 3 shows the calculated results of the bioclimatic evaluations and traditional dwelling adaptations to the geomorphological and climatic aspects. Section 4 providesspecific strategies through the macro, middle and micro levels, respectively.

## 2. Methodology

### 2.1. Study Area

#### 2.1.1. Location

The Yangtze River Delta region is located in eastern China (Figure 3), comprising three provinces (the Jiangsu Province, Zhejiang Province and Anhui Province) and the Shanghai municipality [33]. It has been an important region of the Chinese economy, society and culture since ancient times, with rapid urbanization in recent decades. With the rapid growth of the urban population, sustainable development and the use of green construction land resources are required.

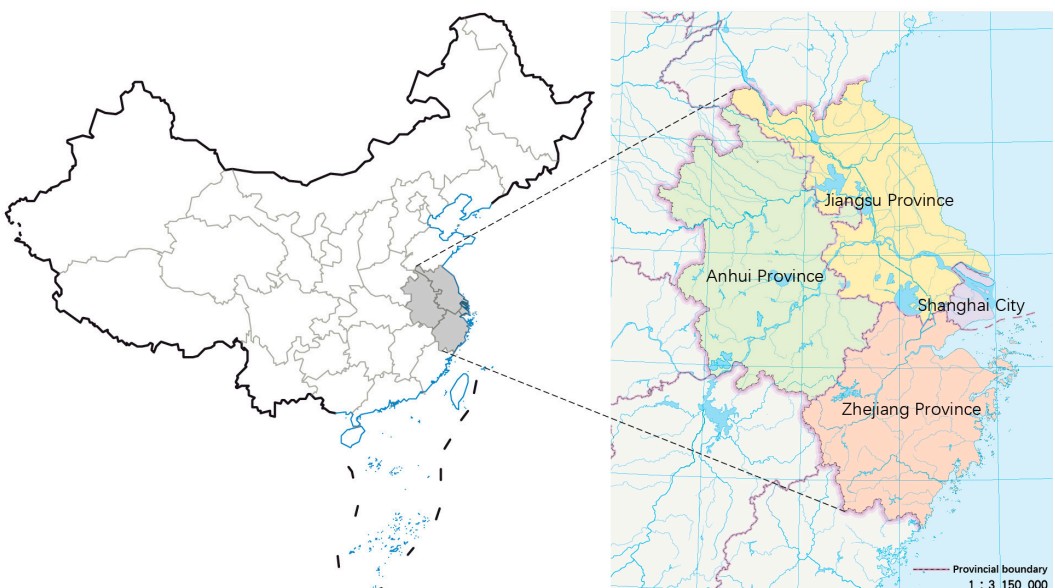

**Figure 3.** Location of the Yangtze River Delta region, China.

As it is located in China's hot-summer and cold-winter zone [34], the regional climate causes the indoor environmental quality of buildings to be far lower than in the rest of China, leading to multiple demands for building cooling, heating, dehumidification and ventilation. Therefore, it is of great pertinence and practical significance to conduct green building strategies based on the geomorphologic and climatic investigations in the Yangtze River Delta.

#### 2.1.2. Geomorphology

The landform in the Yangtze River Delta is rich in patterns, elevations and unit forms, where mountains and hills dominate the west and south while water networks in

plains dominate the northeast (Figure A1). There are three geomorphological types in the study area (Table 1). The overall terrain slopes from southwest to northeast are a unique characteristic of the fragmented landforms, where plain areas comprise the main part.

**Table 1.** Basic geomorphological types in the Yangtze River Delta region.

| Types | Characteristics |
|---|---|
| Mountains and hills | • Geology is unstable, which causes soil movement, such as collapse and landslide, due to the change in external factors such as climate.<br>• Complex terrain with significant height differences causes apparent fluctuation.<br>• Rich vegetation functions in soil and water conservation and the regulation of the microclimate. |
| Plains with river networks | • Terrain is gentle with a slight fluctuation and few high mountains.<br>• Rich water system mainly shows mesh distribution, leading to the fragmented landforms.<br>• Dynamic nature of hydrology and the persistence of flow may cause the development and change of the local geomorphology. |
| Coastal area with island | • Coast twists with the fragmented landscapes.<br>• Homogeneity of the climate, resources and biological population inside the island is high.<br>• Ecological environment has an intense fragility and variability. |

Though varied in geomorphology, the early settlements in the Yangtze River Delta region mainly originated in plain areas or gentle slope areas with sufficient water sources. Therefore, most urban construction occurred in the plains areas.

### 2.1.3. Climate

The Yangtze River Delta usually shares similar climatic conditions in different cities. It has extreme temperatures in the summer, the maximum of which once reached over 40 °C, while in the winter, the temperature averages around 0 °C [35] (Figure 4). Since the temperature ranges on a wide scale, it requires particular approaches to deal with the uncomfortable climates, especially in the summer. In addition, the annual average relative humidity is as high as 80%, and the wind at night is constantly slow and moderate, with a high static annual wind rate.

### 2.1.4. Vernacular Dwellings

A conventional dwelling is considered a living organism from a long-term adaptation to the natural conditions [2,37]. It has evolved through time and reached its optimized condition through trial and error, generally appearing in typological patterns of building organizations, envelopes, spatial layouts, etc. [38].

Vernacular dwellings in the Yangtze River Delta region (Figure 5) have developed a series of conventional strategies for environmental adaptations for five aspects: thermal comfort, solar radiation, humidity, ventilation and natural lighting [39]. To counteract climate change, especially extreme temperatures and climatic disasters, passive adaptation measures and the solar energy utilization of dwellings have shown potential superiority [40]. They respond to the needs of their habitats, climatic conditions and geomorphologic characteristics, providing a reference for the strategies of contemporary green building. Dwelling responsive behaviors to the geomorphology and climate are based on

passive design principles that could be adapted to current green building design practices to optimize the relationship between humans, buildings and the environment.

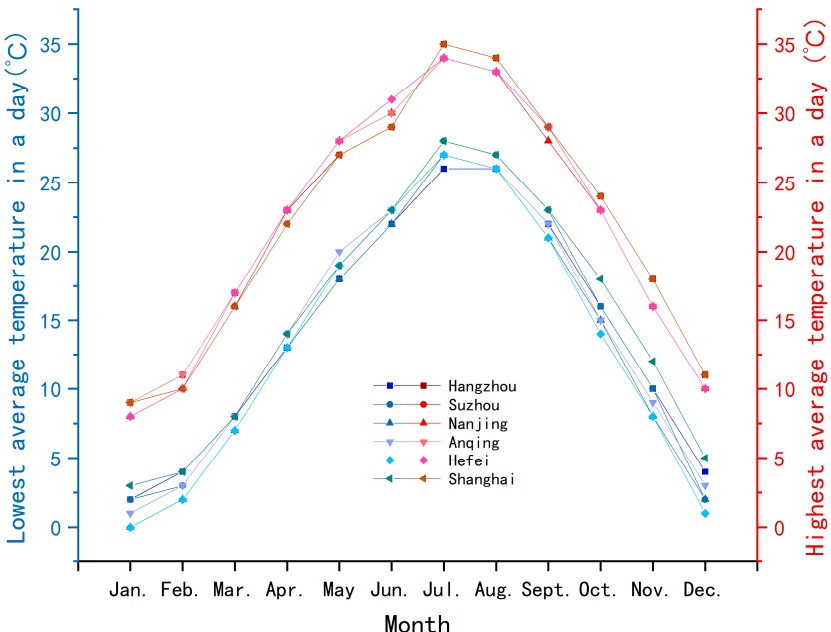

**Figure 4.** The monthly average temperature in a day (2009–2018) in six cities of the Yangzte River Delta region (data source: [36]).

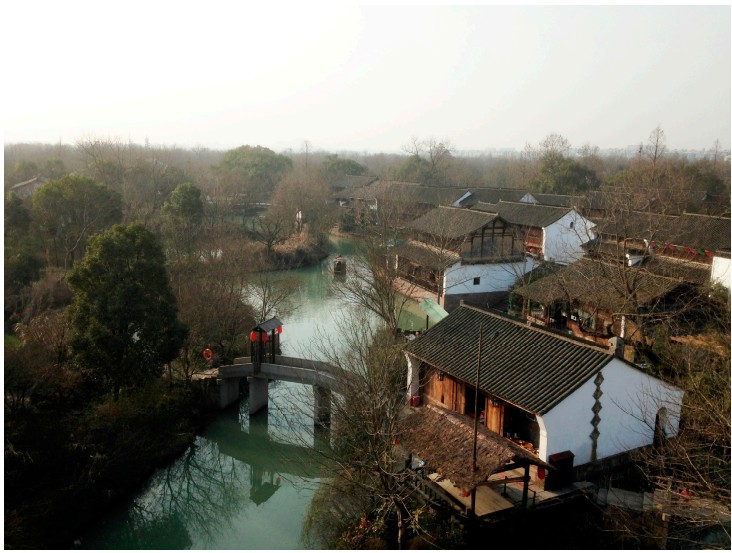

**Figure 5.** Conventional dwellings in the Yangtze River Delta region (photograph taken on 15 January 2016, at Hangzhou Xixi National Wetland Park, Zhejiang, China).

*2.2. Research Methods*

The research framework involved two steps, First, we evaluated six common passive design strategies—preset in the 'Weather Tool' software [41]—based on local climatic data. Second, we learned from the past by recognizing the architectural prototypes in the vernacular dwellings. Specific green building design strategies were developed based on the pre-evaluation of the bioclimatic calculation and the typological interpretation of the vernacular dwellings (Figure 6).

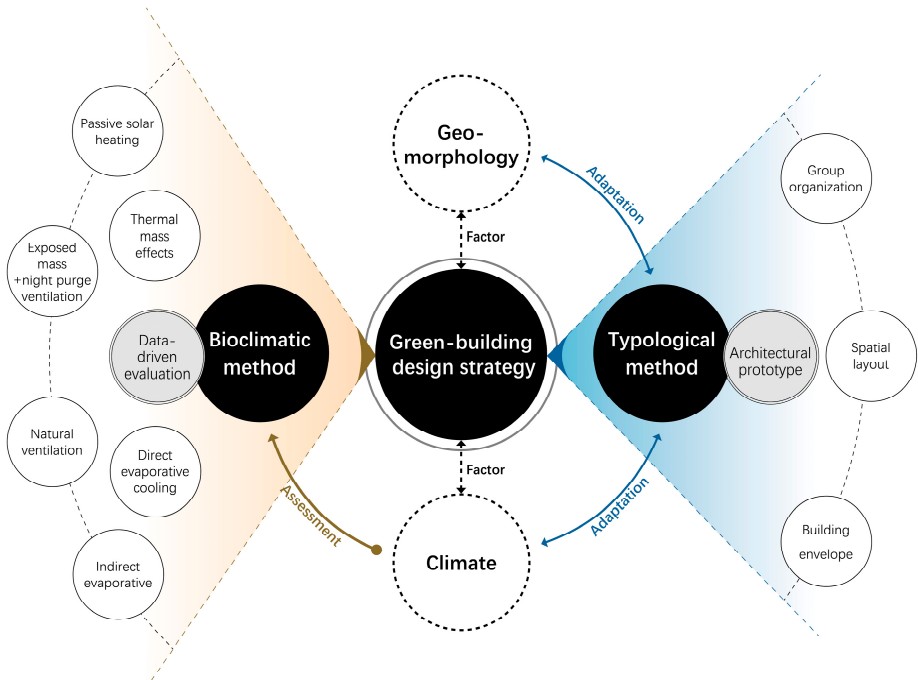

**Figure 6.** Developing green building design strategies based on bioclimatic and typological methods.

### 2.2.1. Bioclimatic Evaluations

In this study, we selected six cities representative of the Yangzte River Delta region: Hefei (31.9° N, 117.2° E), Anqing (30.5° N, 117.1° E), Shanghai (31.4° N, 121.4° E), Nanjing (32.8° N, 118.8° E), Suzhou (34.3° N, 117.2° E) and Hangzhou (30.2° N, 120.2° E). They are either provincial capitals or large-population cities (Figure A1). The 'Weather Tool' was used for the bioclimatic evaluations by inputting the local climatic data, where the software preset the six common passive design strategies: passive solar heating, thermal mass effects, exposed mass and night purge ventilation, natural ventilation, direct evaporative cooling and indirect evaporative cooling. The input data were detailed as follows.

- The climatic data came from the Typical Meteorological Year (TMY) weather files, including the dry and wet bulb temperature, relative humidity, air pressure, wind speed and direction, etc. The TMY means a year with typical climate characteristics in a certain region extracted from long-term and continuous meteorological records. It is the preferred outdoor meteorological design condition for the dynamic analysis of building energy efficiency, simulating building energy consumption and passive building design [42].
- The data format used in this study was the CSWD (Chinese Standard Weather Data), which is a special meteorological data set for building a thermal environment analysis in China developed by the Meteorological Reference Room of China Meteorological Information Center and the Department of Building Technology of Tsinghua University. This database collected the measured meteorological data of 270 ground meteorological stations in China from 1971 to 2003 [43].

The effective time ratio [42] was the major outcome calculated in the 'Weather Tool' for the six passive strategies in particular cities. Specifically, the higher the ratios, the longer the time for improving human comfort in the indoor environment. After calculation, the ArcGIS tool visualized the geographical distributions of the bioclimatic performance of the six passive design strategies in the Yangzte River Delta region. This allowed us to select the effective strategies according to the time ratios.

2.2.2. Architectural Typology

The analysis of the conventional sustainability of the built environment in the Yangtze River Delta region was based on long-term fieldwork, including structural surveys and interviews with inhabitants, local designers and academic experts. Based on the bioclimatic evaluation results, we identified the architectural typology in response to the local geomorphology and the climate characters, from which we could obtain conventional wisdom. Then, we transformed them through the architectural pattern languages for adaptive design [44,45].

The dwellings in the Yangtze River Delta demonstrate the localized characteristics in the urban morphology, building structure and humanistic aesthetics with the physical and spatial patterns. The typology defined the most distinguishing forms that played a dominant role in the regional construction. We were mainly concerned with the built environment through three major aspects (Figure 7)—group organization, spatial layout and building envelopes—to explore their adaptive correlation with the geomorphology (landform, river, elevation, vegetation, etc.) and climate (temperature, wind, humidity, solar radiation, etc.) in the macro, middle and micro levels.

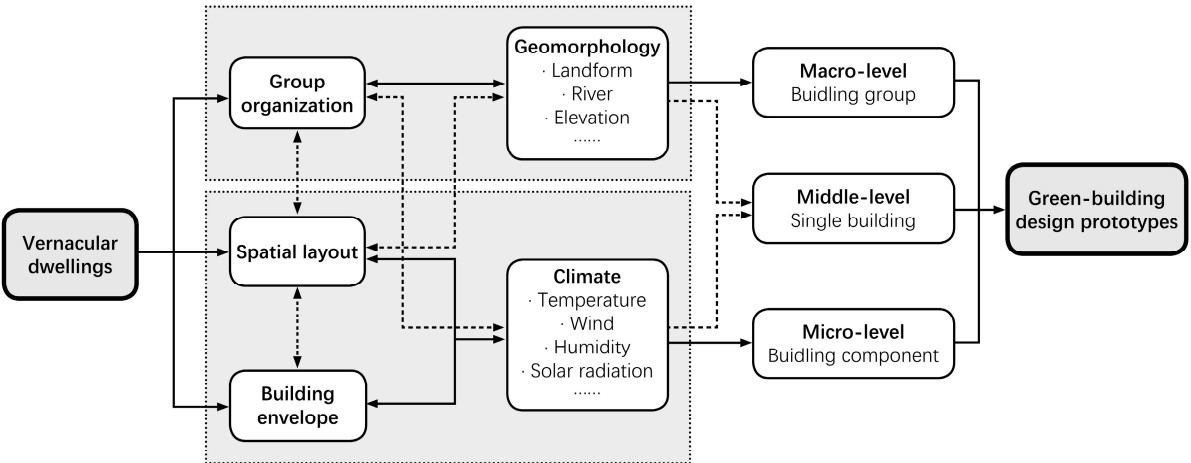

**Figure 7.** Learning from the conventional paradigm by interpreting the architectural prototype.

### 3. Results

*3.1. Calculation Results for Pre-Evaluation*

Figure 8 shows the bioclimatic results of the six cities in the Yangtze River Delta region. It was hypothesized that there were Four passive design strategies—natural ventilation, thermal mass effects, indirect evaporative cooling and passive solar heating—which were appropriate in the Yangtze River Delta region. Additionally, it indicated that the results of the bioclimatic calculation among the different cities in the Yangtze River Delta region were roughly similar. Therefore, the following strategy could be commonly used in the Yangtze River Delta region.

Figure 9 shows the geographical distribution by ArcGIS for the presentation of the results.

- The effective time ratios for the natural ventilation increased from north to south, while the ratios for thermal mass with or without night ventilation gradually decreased from north to south.
- The areas with highly effective time ratios for passive solar heating were mainly located in the east, whereas the areas with small ratios were distributed in the west along mountainous field areas where solar radiation is relatively lower due to the geomorphologic factors.
- Compared to the other five strategies, the effective time ratios of the direct evaporative cooling were relatively small.

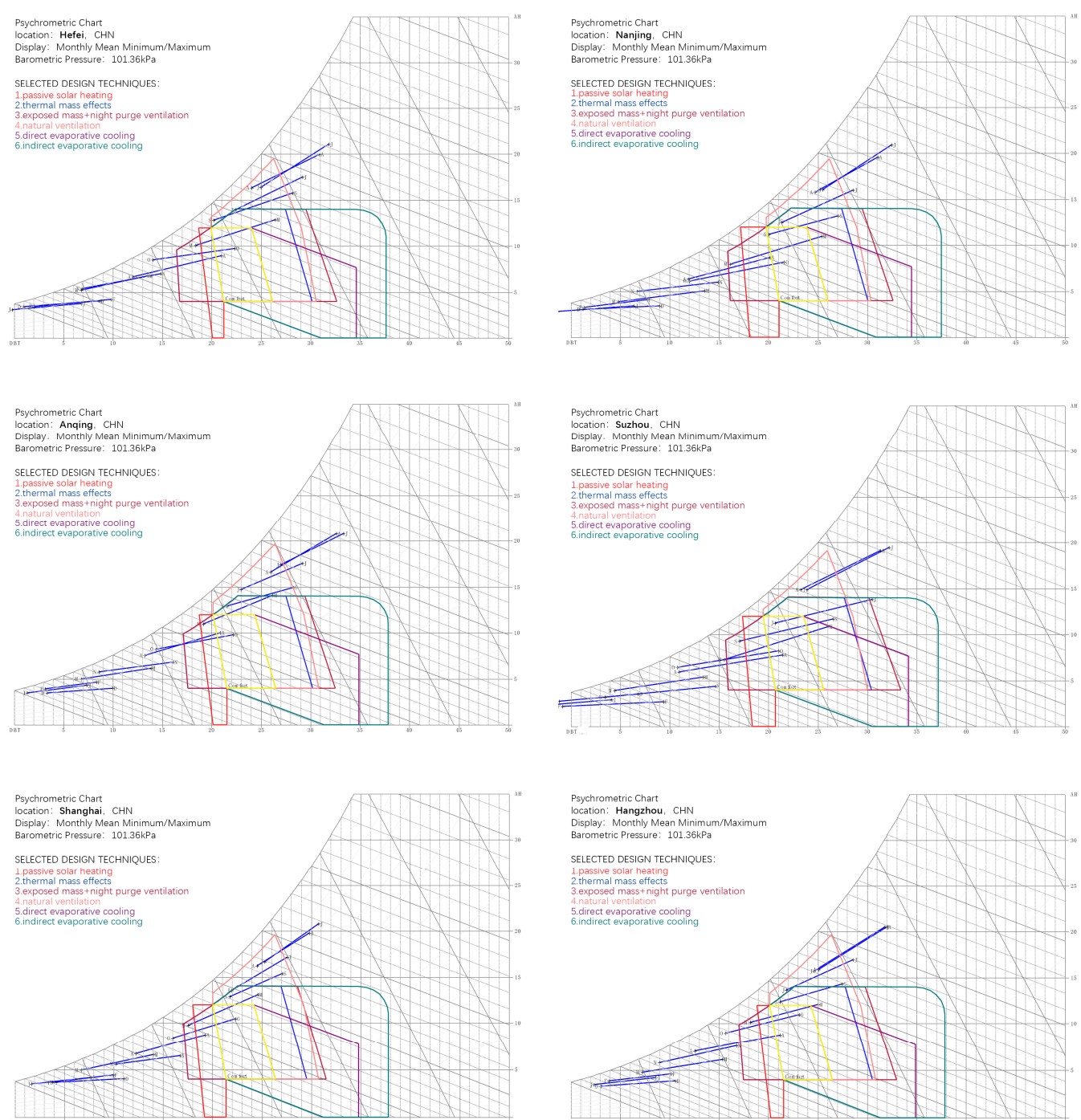

**Figure 8.** Bioclimatic charts of the six cities (Hefei, Anqing, Shanghai, Nanjing, Suzhou and Hangzhou) in the Yangtze River Delta region.

Table 2 shows that the comfortable time ratio without any human-aided regulation measures (comfort zone) in the Yangtze River was about 7.9%, which was only approximately 29 days in total, found mainly in April, May and October. However, the number of comfortable days could be increased to approximately 110 days (accounting for 30% of the whole year) to 130 days in total, by applying the six common passive design strategies comprehensively.

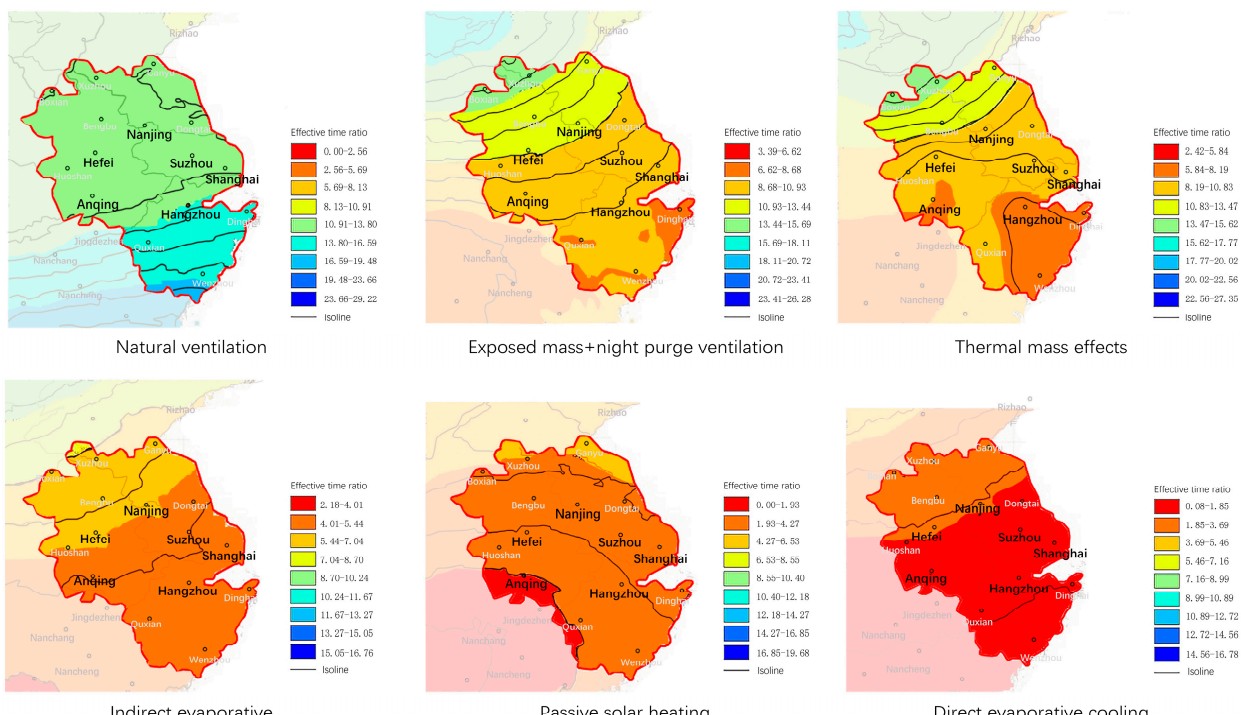

**Figure 9.** Geographic distribution of the six passive design strategies' effective time ratio.

- The natural ventilation had the most significant overall effect throughout the year, and the effective time ratio for natural ventilation was 21%. According to the calculation, thermal comfort could be achieved on 90% of the days between June and September.
- The effective time ratios for the thermal mass with or without night ventilation, indirect evaporative cooling and passive solar heating were 18%, 8% and 6%, respectively.
- The direct evaporative cooling is the more effective passive strategy for hot and dry climate zones and has little efficiency in this region due to its high humidity. Therefore, the direct evaporative cooling measures may not be considered and adopted in this area.

### *3.2. Prototype Results from Conventional Wisdom*

#### 3.2.1. Group Organization

A hierarchical relationship distinguishingly exists in organizing urban spaces and buildings [46]. A room is a primary building cell, designed with an inner courtyard around other rooms. Then, a cluster of building cells forms a group, shaping the form of a block. Finally, the regional settlement is formed (Figure 10). The units can be topologically deformed and connected in multiple directions horizontally and vertically to adapt to complex and diverse landforms [47].

Settlements are developed in response to the sun orientation, wind direction and topography. Their forms and sizes vary according to the location and terrain features (i.e., slope, soil, ground vegetation). With irregular planar forms, they create a comfortable microclimate based on the principle of reducing disturbances to the environment (Figure 11), which also reflects the economical utilization of land resources. Taking vertical site planning as an example, buildings built by the river bank or in mountainous areas circumscribe to the limited site, which sustains the natural surroundings to the maximum extent.

**Table 2.** The effective time ratios of the six passive design strategies in the Yangtze River Delta region.

| | | Month | | | | | | | | | | | | All Year | Days |
|---|---|---|---|---|---|---|---|---|---|---|---|---|---|---|---|
| | | Jan. | Feb. | Mar. | Apr. | May. | Jun. | Jul. | Aug. | Sept. | Oct. | Nov. | Dec. | | |
| | Comfort zone | /[1] | / | / | 7% | 40% | 8% | / | / | 16% | 24% | / | / | 7.9% | 29 |
| | Passive solar heating | / | / | / | 19% | 23% | / | / | / | 4% | 25% | / | / | 6% | 22 |
| | Thermal mass effects | / | / | / | 41% | 51% | 16% | / | / | 54% | 46% | / | / | 18% | 66 |
| Passive design strategy | Exposed mass and night purge ventilation | / | / | / | 41% | 51% | 20% | / | / | 54% | 46% | / | / | 18% | 66 |
| | Natural ventilation | / | / | / | / | 29% | 76% | 27% | 37% | 81% | / | / | / | 21% | 77 |
| | Direct evaporative cooling | / | / | / | / | 5% | / | / | / | 2% | / | / | / | 0.6% | 2 |
| | Indirect evaporative cooling | / | / | / | / | 29% | 21% | / | / | 49% | / | / | / | 8% | 29 |
| | Amount | / | / | / | 41% | 59% | 80% | 27% | 37% | 86% | 46% | / | / | 38% | 139 |

[1] /—No effectiveness.

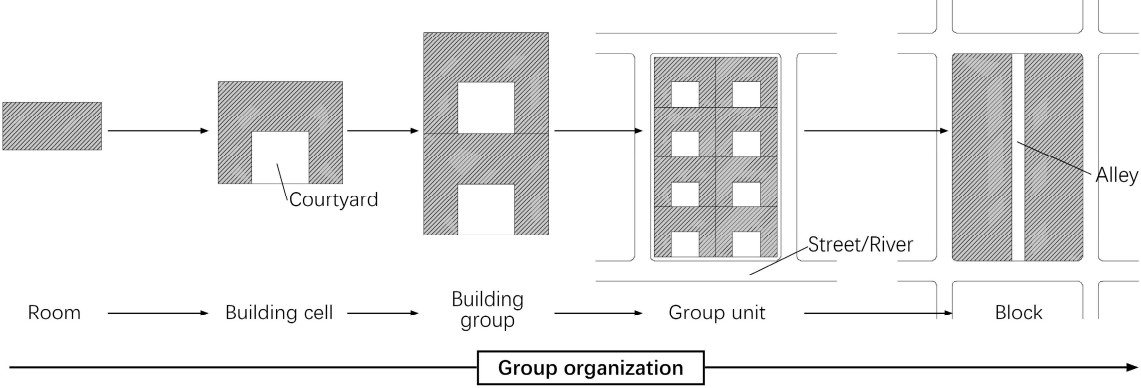

**Figure 10.** Hierarchical building organization principle in the Yangtze River Delta region.

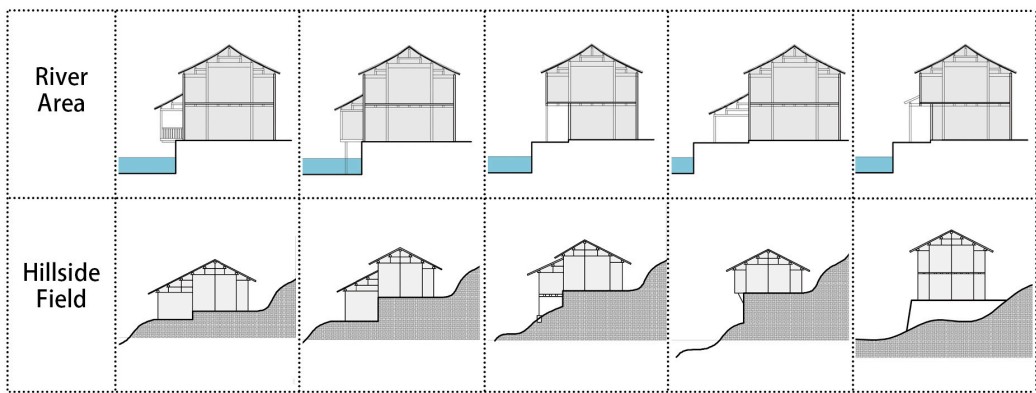

**Figure 11.** Section types of dwellings that have been adapted to different geomorphological forms in the Yangtze River Delta region.

### 3.2.2. Spatial Layout

In most cases, the orientation of the vernacular architecture varies from the 15° northeast axis to the 15° northwest axis, with some flexible orientations found in particular terrains. They face the direction of the dominant wind in summer, as suggested in bioclimatic charts, to maximize the effects of natural ventilation.

Additionally, courtyards serve as an additional climate modifier, ensuring comfort indoors and outdoors (Figure 12). Rooms are provided with sufficient natural ventilation through the appropriate spatial layout to take advantage of the breezes.

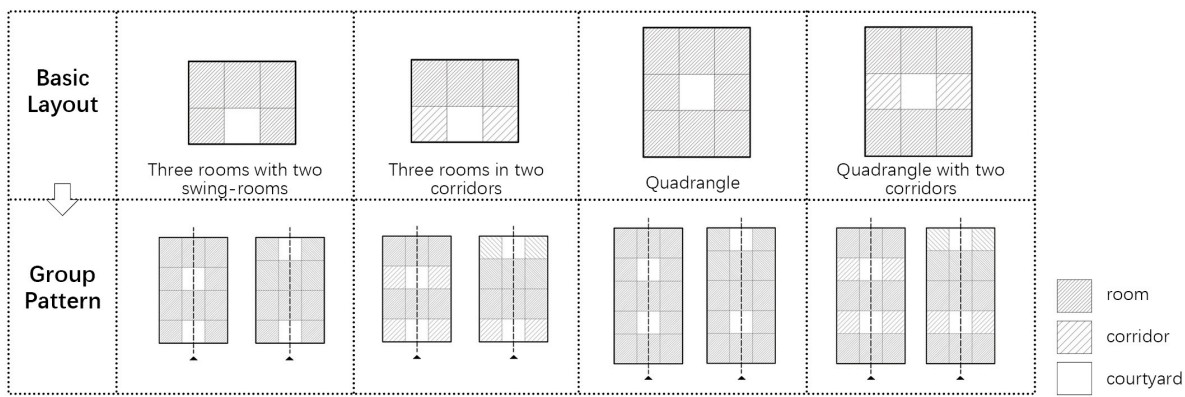

**Figure 12.** Spatial layouts of conventional dwelling groups in the Yangtze River Delta region.

3.2.3. Building Envelopes

Specific details of conventional building envelopes were interpreted through the following aspects: building forms, walls, roofs and windows.

- The shape coefficient [24] of a building denotes the ratio of the exterior area of the building in contact with the outdoor atmosphere to the volume surrounded by it. All vernacular buildings are as compact as possible, which provides the maximum volume with the minimum area exposed to the outside.
- Thick double-layer walls have been most commonly used, aligning with the recommendations in the bioclimatic charts, to improve thermal mass effects. This type of wall section has a clearance of air or soil, which serves as a good insulator. Local materials (e.g., brick, adobe, stone and timber) are employed according to the functional features and site location. The facades' light-colored surfaces are used to protect the walls from solar radiation as they absorb less heat in the summer.
- Influenced by rainfall and solar radiation, most Yangtze River Delta region dwellings were built with sloped roofs. According to survey statistics, the basic slope is 22° to 30°, increasing from north to south. Builders use roof construction materials that possess high thermal capacities. The space between the ceiling and the roof also provides natural ventilation.
- The openings are protected from summer solar radiation by using fixed or moveable shading devices, such as external wooden shutters, so that buildings can be completely shaded in summer but exposed to solar radiation during winter. Such devices are also used to control natural ventilation. Some types of windows can be opened entirely or even removed to increase the width of the indoor air duct to obtain the best natural ventilation and heat dissipation effects in summer.

Additionally, the traditional wooden frames [48] adopted in local dwellings are considered to be the supporting system with modulus characteristics and standardization. The horizontal and vertical distances between each purlin are isometrically arranged. Therefore, the depth, opening, height and space division of the frame and the construction and extension of the building can be flexibly changed, allowing for good adaptability to different landforms.

## 4. Developing Green–Building Design Strategies

The specific green building design strategies in the Yangtze River Delta region were detailed in terms of the macro level (group patterns), middle level (single-building forms) and micro level (building components). These were generated based on the pre-evaluation results of the bioclimatic charts and conventional wisdom by learning from past capacities for adapting to the local geomorphology and climate.

### 4.1. Macro Level: Building Group Aspect

4.1.1. Organization in Groups

Traditional settlements show the wisdom of taking a series of basic building cells to form the pattern for a group of buildings [49]. These spatial cells with square plans can undergo topological deformation, grow, connect and combine flexibly in multiple directions, which has good adaptability to the complex and diverse broken landforms in the Yangtze River Delta region. Prefabricated building systems with the advantages of standardization and modularization, unlike the traditional wooden frames, provide technical support for these cell constructions in contemporary practice [50]. In addition, the concentrated pattern of the building groups reduces the entire settlement shape coefficient, which creates a reciprocal shading for cooling between buildings in summer.

4.1.2. Layout under Hierarchy

The dwellings in the Yangtze River Delta region have usually been built in pieces, forming a particular scale effect [51]. The transition from the outdoor shifting environment

to an indoor comfortable and steady environment is achieved through a set of rooms, corridors and courtyards (Table 3).

**Table 3.** Green–building design strategies at the macro level in the Yangtze River Delta region.

| Environmental Factors | Aspect | Green–Building Design Point | | Evaluation |
|---|---|---|---|---|
| Geomorphology | Settlement organization | (a) | Organized in groups reducing total surface areas | ◆ Passive solar heating [1] |
| | | | | ◆ Thermal mass effects |
| | | (b) | Prefabricated components | ◇ Exposed mass and night purge ventilation |
| | | | | ◆ Natural ventilation |
| | | | | ◇ Direct evaporative cooling |
| Climate | Site planning | (a) | Layout under a hierarchy from outdoor to indoor environment | ◇ Indirect evaporative cooling |

[1.] ◇—ineffective; ◆—effective.

- Rooms were set at the north of the buildings to interact indirectly with the climatic factors behind corridors and a courtyard.
- Corridors acted as public pathways between rooms and as channels for wind ventilation and solar shading.
- A courtyard was directly open to the outdoors as a microclimate container.

*4.2. Middle Level: Single-Building Aspect*

4.2.1. Proper Location and Orientation

The building orientation is usually toward the south to gain sufficient solar radiation and wind ventilation. This deals with urban heat by allowing wind in the summer and sunlight in the winter to enter the indoor space as much as possible. In addition, reducing the spacing between single buildings saves land for construction. In terms of the geomorphology setting, diverse design models based on traditional typology can be used individually or by combining more than one to create a more responsive and sustainable mode, such as excavation, landfill and elevation.

4.2.2. Regular Building Form

An effective strategy for reducing solar heat in summer but preserving heat in winter is to limit a building to a relatively low shape coefficient [52]. Comparing the diverse building plan forms, the more straightforward and regular a plan form, the lower the shape coefficient for the building (Table 4). In addition, considering the climatic conditions of a hot summer and a cold winter, a changeable plan shape was one of the operable solutions, compact in winter to reduce the convective heat exchange while open and stretched when sufficient ventilation is needed in summer. This can be achieved by adjustable envelopes, and similar strategies have been used in current construction and renovation projects [53].

*4.3. Micro Level: Building Component Aspect*

4.3.1. Internally Oriented Room

Though the building form is supposed to be regular and concise, the interior layout can be diverse and flexible. Rooms with different functions are mainly organized around the inner courtyards, which serve as microclimate adjusters for the connected rooms (Figure 13). As suggested in the building bioclimatic charts, natural ventilation is the most significant passive design strategy in the Yangtze River Delta, which can increase comfort by 21% (Table 2). Concerning this region's high static wind rate, the courtyard is conducive to forming a stack effect to promote natural ventilation. In addition, spaces such as balconies,

verandahs and eaves galleries are architectural components that create pronounced shade on the sun-warmed facades and protection from rain exposure.

**Table 4.** Green–building design strategies at the middle level in the Yangtze River Delta region.

| Environmental Factors | Aspect | Green–Building Design Point | Evaluation |
|---|---|---|---|
| Geomorphology | Orientation | (a) Oriented south, southeast, or southwest, mainly within the range of 15° south by east to 15° south by west <br> (b) Remaining consistent with the dominant wind direction in summer in hill areas <br> (c) Built along the riverfront in the plain areas | ♦ Passive solar heating [1] <br> ◊ Thermal mass effects <br> ◊ Exposed mass and night purge ventilation <br> ♦ Natural ventilation <br> ◊ Direct evaporative cooling <br> ◊ Indirect evaporative cooling |
| Climate | Building Form | (a) Simple and regular forms <br> (b) Connecting buildings by sharing walls <br> (c) Rectangular planform <br> (d) Multiple floors <br> (e) Changeable by adjusting envelopes | ◊ Passive solar heating <br> ♦ Thermal mass effects <br> ♦ Exposed mass and night purge ventilation <br> ♦ Natural ventilation <br> ◊ Direct evaporative cooling <br> ◊ Indirect evaporative cooling |

[1] ◊—ineffective; ♦—effective.

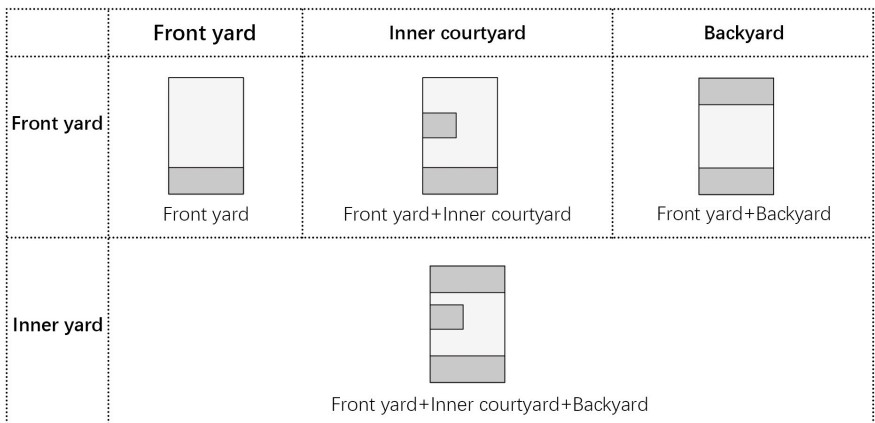

**Figure 13.** Diverse positions of the inner courtyards on the building plan.

4.3.2. Detailed Building Envelope Construction

The building envelopes were analyzed through the vertical elevations—walls and windows—and the horizontal elements—roofs and floors (Table 5).

- The use of building envelope materials with high thermal capacity conforms to the design recommendations in the bioclimatic charts to improve the thermal mass effects. Veneer walls with sandwich insulation at 0.38~0.7 W/(m·K) and external insulation at 0.24~1.0 W/(m·K) [54] have a good climate adaptability in this region. The double-layer wall shows a similar effect of reducing the absorption of solar radiation and the surface temperature by combining ventilation design.
- Removable and deformable awnings in the Yangtze River Delta allow for solar radiation to be captured during the cold season and limits this process during summer. Different components are provided, such as a louvre, sunblind and horizontal or vertical slats. A greenhouse adjoined to the windows is a similar strategy that cap-

tures more solar radiation with a moving mechanism that conducts dissipation by natural ventilation.

- For the consideration of drainage and the reduction in solar radiation, a sloping roof is usually adopted. In terms of the material selection, materials of high thermal mass are generally used. The most frequent occurrence is that a roof covered by vegetation or water is charged with heat energy from solar radiation and later emits this energy to their connected rooms [55]. Roofs with double layers also work as a good insulator and favor heat dissipation through cross ventilation [56].
- For wet and rainy climate conditions, raising the foundation and elevating the ground floor are two common methods used to achieve moisture insulation. Additionally, capacitive flooring absorbs solar thermal energy and modulates the interior temperature. Flooring combined with a ground-source heat pump has also been used extensively due to the advantage of the thermal stability of the ground.

**Table 5.** Green–building design strategies at the micro level in the Yangtze River Delta region.

| Environmental Factors | Aspect | Green–Building Design Point | Evaluation |
|---|---|---|---|
| Climate | Spatial Layout | (a) Enclosed external walls<br>(b) Indoor ventilation by wind and thermal pressure<br>(c) Buffer room for climatic interaction | ◇ Passive solar heating [1]<br>♦ Thermal mass effects<br>♦ Exposed mass and night purge ventilation<br>♦ Natural ventilation<br>◇ Direct evaporative cooling<br>◇ Indirect evaporative cooling |
| Climate | Building Envelope — Roof | (a) Pitched roof with 22–30° incline<br>(b) Roof materials of high thermal resistance and thermal inertia<br>(c) Air cavity for heat dissipation | ♦ Passive solar heating<br>♦ Thermal mass effects<br>♦ Exposed mass and night purge ventilation<br>♦ Natural ventilation<br>◇ Direct evaporative cooling<br>♦ Indirect evaporative cooling |
| | Building Envelope — Wall | (a) White wall as solar reflector<br>(b) Wall materials of high thermal resistance and inertia<br>(c) Bricklaying for thermal insulation effect | ♦ Passive solar heating<br>♦ Thermal mass effects<br>◇ Exposed mass and night purge ventilation<br>♦ Natural ventilation<br>◇ Direct evaporative cooling<br>◇ Indirect evaporative cooling |
| | Building Envelope — Floor | (a) Raised foundation<br>(b) Elevated ground floor | ♦ Passive solar heating<br>◇ Thermal mass effects<br>◇ Exposed mass and night purge ventilation<br>♦ Natural ventilation<br>◇ Direct evaporative cooling<br>◇ Indirect evaporative cooling |
| | Building Envelope — Window | (a) Removable<br>(b) Deformable<br>(c) Double layers | ♦ Passive solar heating<br>◇ Thermal mass effects<br>◇ Exposed mass and night purge ventilation<br>♦ Natural ventilation<br>◇ Direct evaporative cooling<br>♦ Indirect evaporative cooling |

[1] ◇—ineffective; ♦—effective.

All of the strategic tips above are illustrated in Figure 14.

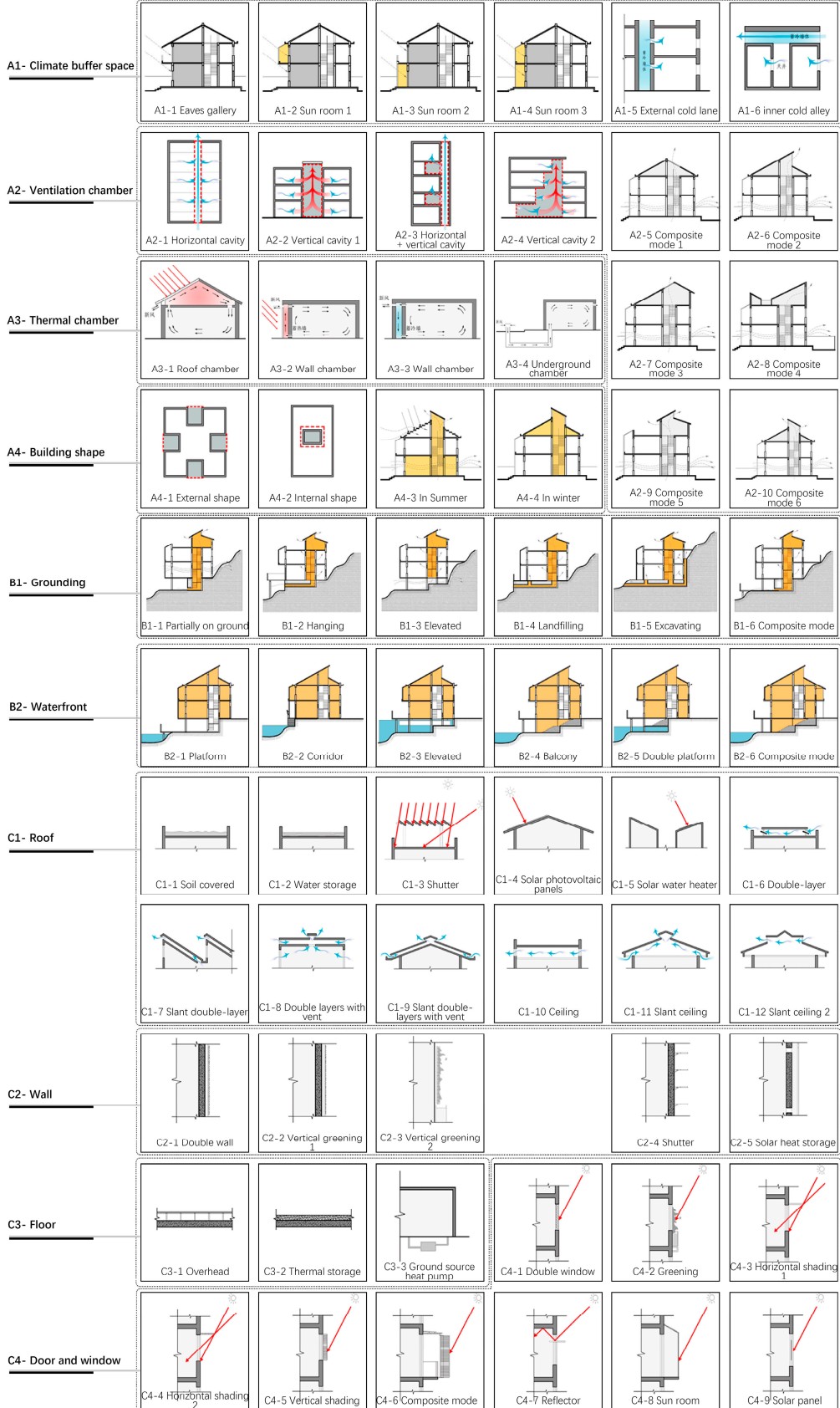

**Figure 14.** Graphic list of the green building design strategies in the Yangzte River Delta region.

## 5. Discussion

Until now, the notion of 'green building' has developed a broad definition, covering environmental, social, economic and cultural sustainability [57]. There are multiple perspectives to investigate green buildings through design, construction, operation, maintenance, renovation and demolition throughout the building life cycle. In this paper, we chose to focus on the early design stage to incorporate the effective strategies for green building development, which aimed to embrace the goals of low energy, low tech and low cost in the long run.

### 5.1. Environmental Adaptations of Design Strategies

Environmentally responsive sustainable design is one of the starting points for green building design and is one of the most significant factors. It is worth noting that green buildings in different countries are designed and built according to the local geographic and climatic conditions, which can also be found in all leading green building assessment tools, such as LEED (United States), BREEAM (United Kingdom) and the Green Mark Scheme (Singapore). There have been several studies reporting these practices [58]. However, the vast majority of the existing studies on green building have focused on the development of green technologies by assessing a building's thermal performance and energy efficiency, which, to a degree, ignores the intensive and dynamic relationships between people, buildings and the natural environment. Some developers tend to amass green technologies blindly for the sake of certifications without considering their suitability for the application in a certain region. Though they indeed play a role in shaping, preserving and improving the built environment, we rather chose to emphasize the importance of low-tech and passive approaches for building design to achieve environmental adaptations.

On the other hand, previous studies had established green building design strategies mainly at the architectural level and paid little attention to multi-scale effects from a systematic perspective. In this study, however, green building design strategies based on architectural typology were interpreted through the macro, middle and micro levels to examine group patterns, single-building forms and building components in a particular locality, comprehensively considering group effects and individual performance.

### 5.2. Pre-Evaluation and Post-Assessment

For realizing the optimization of green building performance, there are obvious advantages to developing a pre-evaluation of suitable strategies in the early stage of green building design. On the other hand, it is possible to measure how the building performance was improved in a post-assessment after the completion of construction compared to the previous conditions if a proper pre-evaluation was applied. The bioclimatic analysis is an effective tool to pre-evaluate the potential of passive design strategies. In this study, according to the effective time ratios, the appropriate strategies in the Yangtze River Delta region were ranked as follows: natural ventilation (21%), thermal mass effects (18%), indirect evaporative cooling (8%) and passive solar heating (6%), which provided the preliminary design suggestions.

Moreover, as listed in the construction menu in Figure 12, every building component has an individual effect from interacting with particular environmental factors, while the building performance is the final result of the components' combined effects. Therefore, we concede that a more precise approach is necessary to access the significant influencing variables in a climate-changing context through simulation and qualitative evaluation [59]. As our findings are based on the current geomorphologic and climatic conditions, this may not be sufficient, as the conditions may change in the future. Therefore, adaptive design strategies for green buildings need to consider future-proofing. More studies are required to validate the real performance of green building strategies via post-occupancy evaluation (POE). Additionally, a user survey on the indoor environment and energy consumption is another effective tool. Those optimizations contribute to achieving a superior design scheme, narrowing the gap between a particular building model designed with multiple

strategies and its building performance, consisting of its indoor environment quality and energy efficiency in the real world. However, this is difficult for architects and decision-makers to simulate or predict.

## 6. Conclusions

Green–building design plays a role on multiple scales and in multiple aspects in response to the regional environment. In this study, we investigated a design approach for generating environmentally-adaptive strategies for green building development in the Yangtze River Delta region. We focused on two natural factors—geomorphology and climate—in the built environment for a deep insight into the human–land system. Through bioclimatic calculation, we evaluated the effectiveness of six mainstream passive design strategies. Then, we extracted the architectural prototypes of vernacular dwellings through three aspects: group organization, spatial layout and building envelopes. Consequently, the specific green building design strategies in the Yangtze River Delta region were based on a combination of the existing strategies and conventional wisdom.

The bioclimatic charts contributed to evaluating the environmentally responsive and sustainable design strategies, which assists architects in realizing the optimization of green building performance. At the same time, learning from the past is beneficial for preserving tacit wisdom and humanistic regionalism [60]. Therefore, we proposed a design approach for integrating both methods, in order to strengthen the effectiveness and adaptations of green building design strategies.

Although the environmental factors and strategies vary worldwide, this study provides insight into the typical morphological and climatic conditions in the Yangtze River Delta region. Based on the local vernacular features, the research framework could be applied to the other green building design scenarios. This study could help support endeavors toward adaptive sustainable development and offer theoretical and practical strategies for architects, designers and scholars. For future work, more details about the strategies should be further verified in architectural practices in the region.

**Author Contributions:** Conceptualization, Y.Z. and Y.S.; methodology, Y.S. and Z.W.; software, Y.Z.; validation, Y.Z., Y.S., Z.W. and F.L.; formal analysis, Y.S.; investigation, Y.S.; resources, Y.Z. and Z.W.; data curation, Y.Z.; writing—original draft preparation, Y.Z.; writing—review and editing, Y.S.; visualization, Y.Z., Y.S. and F.L.; supervision, Y.Z. and Z.W.; project administration, Y.Z.; funding acquisition, Y.Z. and Z.W. All authors have read and agreed to the published version of the manuscript.

**Funding:** This research was funded by the National Key Research and Development Program of China (No. 2017YFC0702504), the Talent Introduction Research Project of Fuzhou University (No. XRC-22010) and the Education and Scientific Research Project for Young and Middle-Aged Teachers of Fujian Province (Science and Technology) (No. JAT210008).

**Data Availability Statement:** Not applicable.

**Acknowledgments:** We thank the editors and reviewers for their kind and valuable suggestions.

**Conflicts of Interest:** The authors declare no conflict of interest.

**Appendix A**

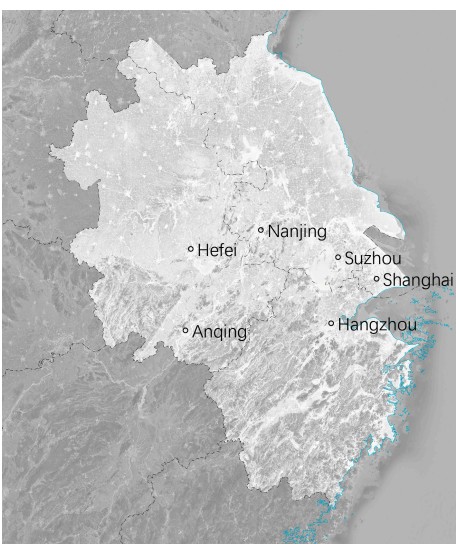

**Figure A1.** General geomorphology of the Yangtze River Delta region, China (satellite image taken by Google Earth on 14 December 2015).

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
