# Peer review of "Developing Green–Building Design Strategies in the Yangtze River Delta, China through a Coupling Relationship between Geomorphology and Climate"

_land, doi:10.3390/land12010006_

Round 1

Reviewer 1 Report (New Reviewer)

Interesting topic worth of continuation, especially in context of longer time (extremal changes of the climat), other regions in the country, continent and the world.

Proper structure of scientific work, but there is needed clearer description of used elements of another authors methodologies, tools ect., and own elemnts of used methodology. 

Proper references to the research of other scientists. 

Reference nr 2 has mistaken desctription. Anna-Maria is the name , surname of this author is Visilia.

Figure 15 - the drawing raises doubts due to the marking of category A2 with two circuits, and lack of conequence in using cicuit around category C4?

Lack of bolded year in many references.

Author Response

Reviewer 2 Report (New Reviewer)

The article describes the issue of green buildings in the Yangtze River Delta in China in the context of geomorphology and climate. The research presented corresponds to the profile of the journal. In the introduction, the authors present the research background, the subject has a lot of literature, the author's selection shows items related to research. There is no clear definition of the purpose of the study and research theses. The chapter on the methods and materials used in the research is very well written. The process is clear, the research conducted consistently. The results are presented in a clear and specific way. The conclusions are not surprising, but it is good that they are supported by research in the local context. The conclusions presented are important for further research.

Author Response

Reviewer 3 Report (New Reviewer)

The reviewed article is of great value and interest. It meets the requirements for scientific publications. The authors have addressed the current topic of green building design. In the research presented, they were located in the Yangtze River Delta region of China. The structure of the work is not objectionable. This is because it contains all the necessary elements such as a description of the purpose of the research, the research methods and the status of the research. 

In the introduction, the authors carefully and logically presented the subject of their research and the reason for undertaking it.  

According to the reviewer, the subject literature cited in the article is well selected and complete.

The article, in the reviewer's opinion, can be published in the submitted form because it was thoroughly and reliably prepared. 

Author Response

This manuscript is a resubmission of an earlier submission. The following is a list of the peer review reports and author responses from that submission.

Round 1

Reviewer 1 Report

The paper presents an interest topics however the scientific approach has to be improved. The research framework has to be explained in more detail. Sections 3.1 and 3.2 have to be more detailed.  Page 11, line 290 - the unit for the k-value is missing Page 11, lines 302-303 - it is not clear what is meant with green roof and how it works. Most of the examples given in Figure 9 are related to heat dissipation through ventilated roofs rather than heat transfer from roof to the room.

Reviewer 2 Report

Thank you for inviting me to review the manuscript “Green Building Design Strategies in the Yangtze River Delta: Interpretation through a Coupling Relationship between Geomorphology and Climate” which investigated how vernacular architecture adapted to local topographical and climatic conditions in the Yangtze River Delta region.

After reading the MS, I have some major concerns related to the problem definition, methodological approach, and scientific contribution of the work as follows:

1.       Problem definition: Authors study “Green building” but the definition is very narrow and biased. Authors define it as “considering local climatic conditions in the design and construction stage” (lines 62-63). This is a narrow consideration since it should take into account not only environmental conditions but also environmental impacts, health conditions, and return on investment. The authors can see variety of definitions from Zuo and Zhao, 2014 (https://doi.org/10.1016/j.rser.2013.10.021). It should also consider in all phases from planning to design, construction, operation, maintenance, renovation, and demolition but not only the design and construction stage as the authors mentioned. Thus, the narrow definition of green building leads to bias throughout the paper.

2.       Methodological approach: The focus of this paper is the relationship between vernacular architecture geomorphology and climate but the description of these variables in the local context were not clear at all (line 108 - 126). Regarding the geomorphology setting, what is the typology of dwellings in each geomorphological type? This help to identify the relationship between building typology and geomorphology. Regarding the climate condition, how about the condition in the winter? It will provide opportunities to study not only the cooling effect in the summer but also the heating effect in the winter. Importantly, there is not any specific data to support the study of the relationship between vernacular architecture and local topographical and climatic conditions. Since the local context is not clear, the proposed strategies of green building design based on architectural typology are not convincing.

3.       The scientific contribution of the work to the scientific community is very weak and unclear. The authors should enlarge the topic and support the theoretical analysis with much more data, at least some comparison with the reality in the local context. In general, the conclusion is very weak and does not support the results of the author’s study. Also, some conclusions are not supported by the results of the author’s work (e.g., t shows little dependence on high technologies, cost, and maintenance but largely on the building itself – lines 338 – 339). Where does this conclusion come from? What is the final message from your conclusion to readers? 

Reviewer 3 Report

Overall, the topic is interesting but the structure and the content of the manuscript should be improved significantly.

Abstract:

The abstract is informative but needs some improvement:

Line 14: not everyone knows where the Yangtze River Delta is. Please add the country after the name of the Delta.

Introduction:

The introduction needs some improvement to make it more informative and understandable for the reader. Here are a few changes that I suggest:

Line 31: I would use etc. rather than et al.

Figure 1: It is better to discuss your conceptual model in one paragraph to provide some information for the reader.

Lines 85-90: discuss in further detail and elaborate on what the authors want to accomplish in this research.

Methodology:

The information provided about the study area is sufficient and informative.

The same issue as before with figure 5. There is not enough information in the text about the research framework. This information needs to be added to the manuscript (the last paragraph of section 2 needs to be expanded to include more information about the research method)

Figure 6 is very informative. However, I am not sure if it is developed by authors or is from other sources. If it is from other sources, the citation is missing.

I couldn’t see this pattern (group organization) in figure 4. The authors might need to add a picture that is showing this group organization in reality.

Section 4 is providing good information on the type of buildings and the specifications. However, the socio-economic aspects should also be discussed briefly, even if they are not the main focus of the study.

Discussion:

The discussion is the weakest part of this study. This section needs to be improved significantly.

Again, there is a chart in the discussion with a one-sentence discussion about it. It is not usually common to bring a new diagram into the discussion especially when there is not enough discussion about it.

The findings and suggestions are not discussed in this section. In other words, the main purpose of the discussion section is ignored.

Although it is good to acknowledge that you are aware of other factors that should be considered in developing contemporary settlements, the reader should be able to understand this from your manuscript before they even get to the discussion section.

Conclusion:

The main components of a conclusion are missing and the conclusion is not providing any wrap-up information on the manuscript (at least the purpose, findings, limitations, and future studies should be discussed in this section)

Reviewer 4 Report

This research employed architectural typology theory to analyze how vernacular architecture adapted to local topographical and climatic conditions in the built environment to take a deep insight into the human-land system in the Yangtze River Delta region. The research themes are of significance and are interesting in green building design associated with built environment. The manuscript has well been organized and acquires some significant conclusions. Some issues still need be addressed before publication.

(1) In the manuscript, green design strategies were interpreted through the macro, middle, and micro levels to examine group patterns, single building forms, and building components in a particular locality. However, the scientific objectives and theory need be strengthened, and methods of assessment is absent.

(2) Please combine the specific case, the assessment of the effectiveness of green design strategies should be added in the manuscript.

(3) The key findings on abstract and conclusion need be rewritten.

(4) The strategies on Nature-Based Solution and climate change adaption should be considered in the revision.

Round 2

Reviewer 2 Report

Thank you for inviting me to review the revised version of the manuscript “Green Building Design Strategies in the Yangtze River Delta, China: Interpretation through a Coupling Relationship between Geomorphology and Climate”. Thank you the authors for the elaboration and revisions to response to my comments. Unfortunately, the changes are minor and they are not sufficient enough. Specifically, the problem definition remains very narrow. The author elaborated a few texts related to geomorphology settings and climate conditions but no further analysis was elaborated to deal with this aspect. The authors did not provide further data analysis and comparison to support the study but state that it is “This study thus contributes to the methodical design field”. The new version of the conclusion is not good at all, especially the idea of elaborating of a figure in the conclusion section. In my point of view, this manuscript fits well as a project report, a deliverable, or the introductory section of a completed research paper. The authors need to spend more effort to collect, elaborate, and analyze more data to turn this work into a publishable paper.

Reviewer 3 Report

The authors have addressed my comments and I believe that the revised version of the manuscript is publishable.

Reviewer 4 Report

The manuscript has been improved in this version. However, article has serious flaws, additional experiments needed, research not conducted correctly. The scientific contribution of the work to the scientific community is very weak and unclear. I can not acquire the significant advance from this research in the revised version.